# Anatomical Classification for Plantaris Tendon Insertion and Its Clinical Implications: A Cadaveric Study

**DOI:** 10.3390/ijerph19105795

**Published:** 2022-05-10

**Authors:** Jeong-Hyun Park, Jaeho Cho, Digud Kim, Hyung-Wook Kwon, Mijeong Lee, Yu-Jin Choi, Kwan Hyun Yoon, Kwang-Rak Park

**Affiliations:** 1Department of Anatomy & Cell Biology, School of Medicine, Kangwon National University, Chuncheon 24341, Korea; jhpark@kangwon.ac.kr (J.-H.P.); oe5235@naver.com (D.K.); kwenhw@naver.com (H.-W.K.); toff337@hanmail.net (M.L.); police5565@hanmail.net (Y.-J.C.); 2Department of Orthopaedic Surgery, Chuncheon Sacred Heart Hospital, Hallym University, Chuncheon 24253, Korea; hohotoy@nate.com; 3Division in Biomedical Art, Incheon Catholic University Graduate School, Incheon 21987, Korea; artanato@naver.com; 4Department of Anatomy, School of Medicine, Keimyung University, Daegu 42601, Korea

**Keywords:** clinical anatomy, anatomical variation, classification, plantaris tendon, Achilles tendinopathy, cadaveric study

## Abstract

The purposes of this study were to ascertain the morphological characteristics of a plantaris tendon (PT) insertion using a larger-scale dissection of Korean cadavers and to classify the types of PT insertion related to the calcaneal tendon (CT). A total of 108 feet from adult formalin-fixed cadavers (34 males, 20 females) were dissected. The morphological characteristics and measurements of the PT insertion were evaluated. Five types of PT insertion were classified, wherein the most common type was Type 1 (39 feet, 63.1%). Type 2 and Type 3 were similar, with 16 feet (14.8%) and 15 feet (13.9%), respectively. Type 4 (6 feet, 5.6%) was the rarest type, and Type 5 had 25 feet (23.1%). The case of an absent PT was noted in 7 feet (6.5%). In the proximal portion, the tendon had a thick and narrow shape, became thin and wide in the middle portion, and then changed to thick and narrow again just before the insertion into the calcaneal tuberosity. This study confirmed the five types according to the location of the PT and the area of its insertion-related CT. The morphology of the PT insertion may be anatomically likely to influence the occurrence of tendinopathy in the CT.

## 1. Introduction

The plantaris muscle (PM) is a small muscle that forms a superficial group of muscles in the posterior compartment of lower leg, and is composed of a spindle-shaped short belly and a long slender tendon. It is also called the triceps surae muscle, along with the gastrocnemius muscle and soleus muscle [1]. The PM usually originates from the epicondyle below the lateral supracondylar line of the femur. The belly is located in the medial, slightly deep portion of the lateral head of the gastrocnemius. It travels along the lateral of the popliteal vessels and tibial nerve and turns into tendons [2]. The tendon passes across the posterior of the soleus and anterior of the gastrocnemius, running distal to the leg [3,4]. The tendon passing between the two muscles runs along the medial border of the calcaneal (Achilles) tendon (CT) from the middle part and is finally inserted into the calcaneus [1].

The PM is innervated by the tibial nerve (S1, S2). The PM acts as a weak flexor of the knee joint and plantar flexor of the ankle joint. As the plantaris tendon (PT) is long and thin, it can be easily confused with the nerve during the dissection of the lower leg. That is why the terms “freshman’s nerve” and “fool’s nerve” were coined in consideration of the common mistakes made by inexperienced medical students [5].

Simpson et al. [6] argued that the PM was an accessory muscle and vestigial organ in humans, and that it may be absent in about 7–20% of the population. However, Menton et al. [7] argued that the PM has a function of finely adjusting the joint motion by constantly and accurately feeding back tension information whilst moving together with the surrounding large muscles, and that this should be regarded as a proprioceptive organ.

The prevalence of CT disorders, including tendinopathy, is increasing [8,9,10]. It is most commonly affected by the mid-portion of the tendons, accounting for 55–65% of CT-related pathology, followed by insertional tendinopathy, which accounts for 20–25% [8,11,12]. Achilles tendinopathy occurs mainly in the mid-portion of the CT and is, generally, a difficult disease to cure. Nevertheless, the pathological mechanisms associated with this disease have not been fully explored [11,13,14].

Recently, increasing attention has been paid to the potential involvement of the PT in relation to tendinopathy occurring in the mid-portion of the CT [13,15,16]. Ahmed et al. [17] discussed that the PM is a structure with a very high morphological variability, and a high degree of understanding of its role and function is required for a clinical application. Accordingly, the need for various and extensive information on the anatomical characteristics of the PT is emerging.

Therefore, the current study was undertaken to ascertain the morphological characteristics of a PT insertion using a larger-scale dissection of Korean cadavers and to classify the types of PT insertion related to the CT. The results of this study provide a reference to better understand the function of the PT through more detailed and extensive knowledge that helps us to understand Achilles tendinopathy with both the insertional and mid-portion of the CT.

## 2. Materials and Methods

All cadavers used in this study were donated through the body donation program with consent for education and research. In addition, the Institutional Ethics Committee gave approval for conducting this study (Institutional Review Board number: NON2021-001).

In this study, cadavers that were agreed upon and donated to the medical school for the purpose of education and research were used. We dissected both feet of 54 adult cadavers (34 males and 20 females) fixed with formalin. The calf region of all the cadavers was in an intact state without any previous trauma or surgery signs and deformities. Of the 108 lower limbs, 68 limbs (63%) were from males and 40 limbs (37%) were from females. The mean age of the donors at death was 76.6 ± 11.37 years and ranged from 44 to 99.

A pedestal was used with the cadavers to ensure stability in a prone position. Skin incision lines were marked with a pen and the dissection was carefully performed using classical anatomical procedures. The skin incision was set with two horizontal incision lines and one vertical incision. The upper horizontal incision line was 10 cm proximal to the line connecting both epicondyles; the lower horizontal incision line was 5 cm distal to the line connecting the tip of both malleoli. The vertical incision line was a line connecting the middle points of both horizontal incision lines. After skinning, the small saphenous vein and sural nerve were found and removed. The origin of the gastrocnemius and the location of the insertion were checked. The gastrocnemius was cut using surgical scissors at the junction of the medial and lateral heads. After the removal of the gastrocnemius, we located the origin of the PM around the lateral epicondyle of the femur. Starting from the origin and dissecting it in the distal direction, the entire PM and PT were then exposed.

The morphology of the PT was then evaluated. Based on the criteria used in a previously reported classification [15], we classified the types according to the location and distribution area of the insertion to the calcaneal tuberosity.

To measure the width and thickness of the PT, three measurement points were set: point A, the junction of the belly and the tendon; point B, the mid-point of the total length of the tendon; and point C, the point where the tendon begins to extend (Figure 1). The measured values of the width and thickness of each measurement reference point of the PT were used to calculate the average, standard deviation, median, and interquartile range. Each reference point identification and measurement was performed independently by two researchers. Three reference points were identified; two researchers measured once each and the average of those values was adopted as a value to describe the reference point. All values were measured with an electronic digital caliper (Sincon Corporation, Korea) with an error (precision) of 0.03 mm or less.

### Statistical Analysis

The inter-class reliability for all measurements was calculated by the inter-class correlation coefficient (ICC). The data were analyzed using IBM SPSS Statistics version 23.0 for Windows (IBM Co., Armonk, NY, USA), and a *p*-value less than 0.05 was considered to be statistically significant. All analyses were performed using nonparametric methods because our data did not follow the normality assumption. The comparisons between the ethnicity, gender, and side measurements of the type of classification according to the insertion of the PT were analyzed by a generalized estimating equation (GEE). The mean comparison of the PT width and thickness according to the gender was analyzed by a Mann–Whitney U test, and the mean comparison between the left and right was analyzed by a Wilcoxon signed-rank test.

## 3. Results

The inter-class correlation coefficient of all measured values for the reliability analysis was 0.97 for the width of the PT and 0.96 for the thickness. All measurements were higher than the accepted reliability (0.8) and were employed in the study.

The plantaris tendon can be classified into five types based on its morphology, its relation to the CT, and the distal insertion point of the calcaneal tuberosity. Type 1 is a form that expands in a fan shape on the medial side of the CT before the tendon insertion and is inserted into the calcaneal tuberosity. Type 2 is inserted into the calcaneal tuberosity along the medial side of the CT similar to Type 1, but does not take a fan shape. Type 3 is characterized by its insertion into the calcaneus anterior to the CT. Type 4 is a form in which the insertion is fused to the deep crural fascia, not to the calcaneus or calcaneal tuberosity. Type 5 is characterized by a very wide insert surrounding the medial and posterior surface of the CT. A schematic drawing of the five types is shown in Figure 2 and a photograph of a dissected cadaver is shown in Figure 3.

Of the 108 specimens of the Korean population, Type 1 was observed most frequently in 39 lower limbs (36.1%). Type 2 and Type 3 were observed with a similar frequency in 16 (14.8%) and 15 lower limbs (13.9%), respectively. Type 4 was observed with the least frequency in 6 lower limbs (5.6%). Type 5 was observed in 25 lower limbs (23.1%). An absent PT was noted in 7 lower limbs (6.5%). No significant differences in type were observed between males and females (Table 1). There was no significant difference in type between the left and right.

The means of the PT width by the reference point were 2.53 ± 0.92 mm for point A, 2.92 ± 0.90 mm for point B, and 2.61 ± 0.78 mm for point C. The means of the PT thickness by the reference point were 0.86 ± 0.29 mm for point A, 0.52 ± 0.16 mm for point B, and 0.69 ± 0.27 mm for point C (Table 2). To assist in identifying the changes in the thickness and width of the tendons by the reference point, a boxplot was created based on the median, range, and IQR (Figure 4).

## 4. Discussion

A clear understanding of the insertional morphology of the plantaris tendon (PT) in relation to the calcaneal tendon (CT) and calcaneal tuberosity may be important for clinicians who encounter Achilles tendinopathy. In this study, we cited the classification method of Olewnik et al. [15], which classified five types according to the location and distribution area where the PT is inserted into the calcaneal tuberosity. They classified the types of PT inserted into the calcaneal tuberosity in more detail based on the four classifications defined by Cummins et al. [12]. Type 2, which is inserted into the calcaneal tuberosity along the medial border of the CT, but does not take a fan shape, and Type 4, which is inserted into the deep crural fascia rather than being located at the calcaneal tuberosity, were additionally proposed. It was evaluated that it would be appropriate to apply this method to this study as a classification criterion to define the existing type classification in more detail.

In the verified material, and as a result of classifying the types of PT insertion, all five types were identified. Type 1 was found to be the most common and occurred in 39 lower limbs (36.1%). Cummins et al. [12] used 94 lower limbs (47.0%) and Olewnik et al. [15] used 22 lower limbs (44.0%). All showed a similar frequency to the previously reported studies. The second most common type in our classification was Type 5, which occurred in 25 lower limbs (23.1%). In the study of Olewnik et al. [15], Type 5 occurred in 11 lower limbs (22.0%), also showing similar results to previously reported studies. Type 4 had the rarest insertion, occurring in 6 lower limbs (5.6%). The results of Olewnik et al. [15] also showed that the rarest insertion was of Type 4, occurring in 2 lower extremities (4.0%). In a peculiar case, Type 3 of this study occurred in 15 lower limbs (13.9%) whereas in the study of Cummins et al. [12], it occurred in 73 lower limbs (36.5%) and was the second most common form in their results. To discover the possibility of differences in the morphological variation of PT insertions according to race, we compared the incidence of the type of PT insertion in the European population of Olewnik et al. [15] and the results of this study in the Korean population. There was no statistically significant difference (*p* = 0.849) (Table 3).

In this study, the thickness and width were measured at three points to confirm the morphological characteristics of the PT. The reference points were set at the junction of the belly and tendon as the starting point, the mid-point for the total tendon length, and the point where the tendon begins to extend as the ending point. Most previous papers did not reveal the exact location of the thickness or width measurements [18,19,20]. In previous studies that presented the measurement point, the thickness or width were measured at only one point and the most measured case was only at two separate points [15,21]. We posited that the thickness and width would not be constant depending on the measurement point when considering the characteristics of the PT in the shape of a long line. Therefore, we judged that a measurement reference point comprising three points was reasonable to investigate the morphological characteristics of the tendon.

As a result, at point A, the tendon thickness was thick and the width was narrow. At point B, the tendon thickness was thin and the width was wide, but at point C, the tendon pattern became thicker and narrower again. This indicated that the tendon thickness and width of each reference point were not constant. Nevertheless, previous studies on the morphological characteristics of long tendons by location could not be confirmed. In previous papers that only measured at two points, the average of the tendon thickness and width was presented, but there was no explanation for the morphological characteristics of the tendon [21]. Olewnik et al. [15] suggested that specific insertion types and courses in the PT may be involved in the induction of pain in patients with ATT. However, no analysis was made on the shape of the overall thickness and width of the tendon. In particular, the result that the thickness and width of the PT were not constant at each position suggested that there was a possibility of causing friction or other negative effects on the surrounding structure during joint motion. Therefore, further studies are needed on the interaction mechanism with the CT due to the change in thickness and width of the PT.

Sterkenburg et al. [14] reported that the PT was closely related to the CT in 11 of 107 study cases. It was suggested that these two tendons generate opposite forces in a few ankle joint movements, causing traction to the peritendinous tissue at the level of the mid-portion CT, resulting in Achilles tendinopathy. The mechanism, they argued, did not consider the tendinous morphological characteristics (changes in thickness and width by location) suggested in this study. However, if the physical movement between the two tendons that are in contact with each other is considered, not only the course or insertion of the PT, but also the change in the shape of the tendon should be treated as important. In healthy normal person, the Achilles and plantaris tendons, being located in the single paratenon, can glide freely without causing pain [22]. The results of our study showed that the CT and PT coursed together with the same paratenon in most cases, except for Types 3 and 4. This suggests that if the PT is thickened in a pathological state, it is highly likely to cause Achilles mid-portion tendinopathy by acting as a compressive or shearing force on the CT. Additionally, in Type 5, the possibility that the PT may act on Achilles insertional tendinopathy cannot be excluded. However, as tendinopathy is caused by various factors such as inflammatory changes and tendon cytokine expression, it should not be overlooked that the association between the PT and CT should be limited only from an anatomical point of view.

This study was limited by the use of fixed cadavers to evaluate the morphological characteristics of the tendons. Regarding post mortem changes, there may be differences in the measurements taken from a live person and those from a cadaver. Furthermore, the cadavers were limited to those of elderly individuals (mean age 76.6 years) and the presence of tendon pathology may not be fully certain because the corpse specimens donated for research did not provide a past medical history.

## 5. Conclusions

This study classified the morphological characteristics of a PT insertion related to the CT using a larger-scale dissection of Korean cadavers. The morphology of the PT insertion may be anatomically likely to influence the occurrence of tendinopathy in the CT.

## Figures and Tables

**Figure 1 ijerph-19-05795-f001:**
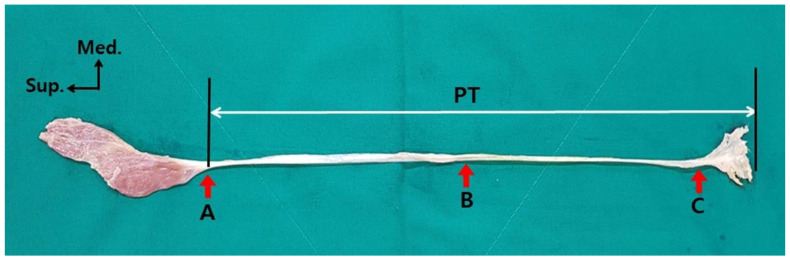
Reference point for measuring the width and thickness of the PT (red arrow): point A is the junction of the belly and tendon, point B is the mid-point of the total tendon length, and point C is the point where the tendon begins to extend. PT: plantaris tendon; Sup.: superior; Med.: medical.

**Figure 2 ijerph-19-05795-f002:**
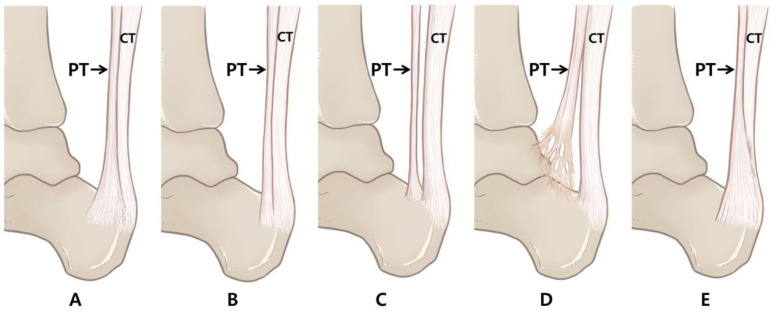
Schematic drawings of the classification of the PT according to insertion type. Medial view of the right leg with the PT. (**A**) Type 1 is a form that expands in a fan shape on the medial side of the CT before tendon insertion and is inserted into the calcaneal tuberosity; (**B**) Type 2 is inserted into the calcaneal tuberosity along the medial side of the CT similar to Type 1, but does not take a fan shape; (**C**) Type 3 is characterized by insertion into the calcaneus anterior to the CT; (**D**) Type 4 is a form in which the insertion is fused to the deep crural fascia, not to the calcaneus or calcaneal tuberosity; (**E**) Type 5 is characterized by a very wide insert surrounding the medial and posterior surface of the CT. CT: calcaneal tendon; PT: plantaris tendon (black arrow indicates plantaris tendon).

**Figure 3 ijerph-19-05795-f003:**
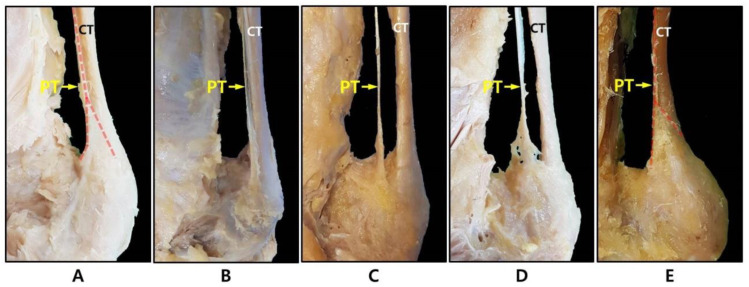
Photo of the classification of the PT according to insertion type. (**A**) Type 1; (**B**) Type 2; (**C**) Type 3; (**D**) Type 4; (**E**) Type 5. CT: calcaneal tendon; PT: plantaris tendon (red dashed line indicates plantaris tendon).

**Figure 4 ijerph-19-05795-f004:**
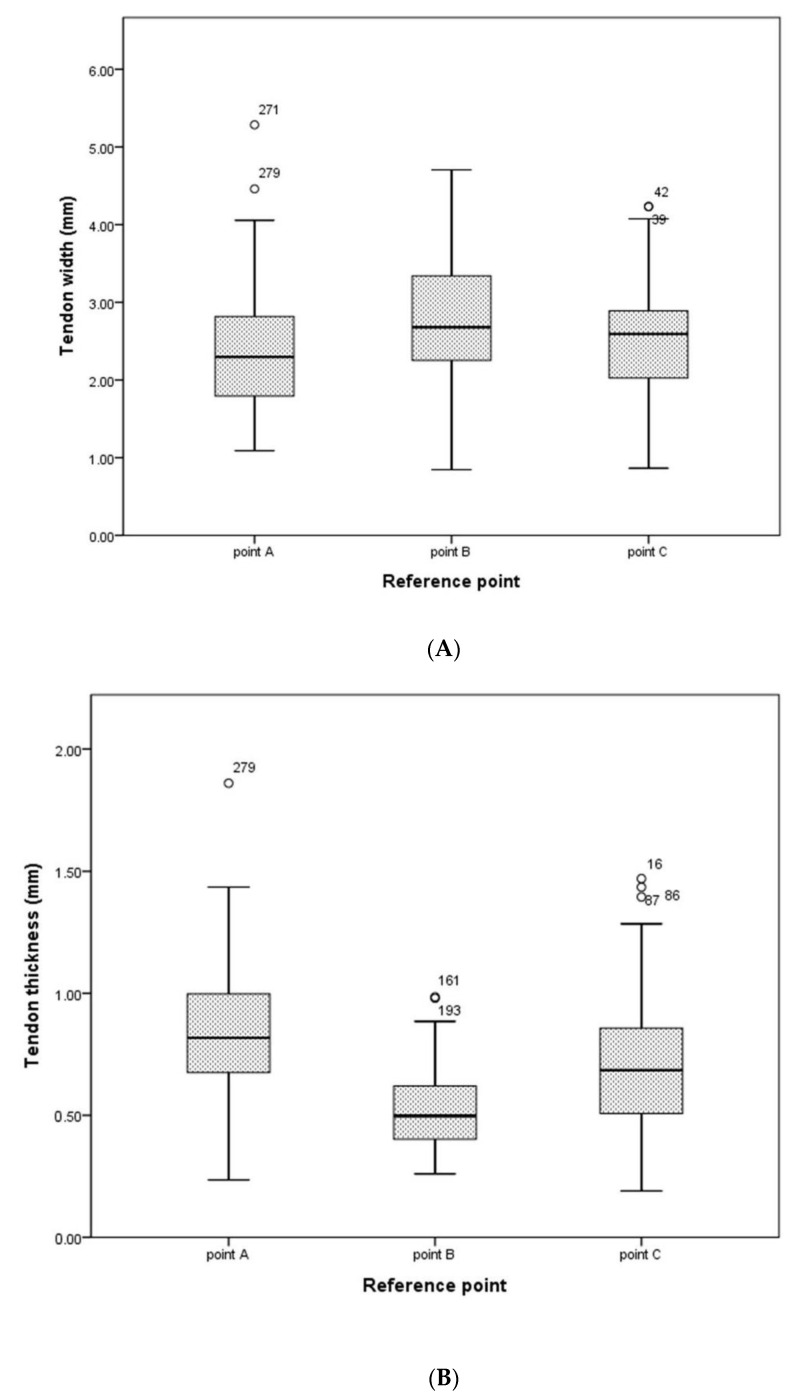
Boxplot of thickness and width of PT for each reference point (*n* = 101). (**A**) Distribution of the width of the PT for each reference point and (**B**) distribution of PT thickness for each reference point. Interquartile ranges are expressed as boxes, the median is the thick horizontal line inside the box, and the range is the horizontal line below and above the box. Point A is the junction point of the tendon and belly, point B is the mid-point for the total tendon length, and point C is the point where the tendon begins to extend.

**Table 1 ijerph-19-05795-t001:** Classification of plantaris tendon insertion type according to gender (*n* = 101).

Type	Female	Male	Total	*p*-Value
**1**	10 (25.0)	29 (42.6)	39 (36.1)	0.289
**2**	8 (20.0)	8 (11.8)	16 (14.8)	
**3**	6 (15.0)	9 (13.2)	15 (13.9)	
**4**	1 (2.5)	5 (7.4)	6 (5.6)	
**5**	11 (27.5)	14 (20.6)	25 (23.1)	
**Absent**	4 (10.0)	3 (4.4)	7 (6.5)	
**Total**	40 (100.0)	68 (100.0)	108 (100.0)	

The data are expressed by number (percent).

**Table 2 ijerph-19-05795-t002:** Median and IQR for width and thickness per reference point (*n* = 101): point A, the junction of the tendon and belly; point B, the mid-point for the total tendon length; and point C, the point where the tendon begins to extend. IQR: interquartile range; Q1: lower quartile; Q3: upper quartile; SD: standard deviation.

Measurement (mm)	Reference Point	Mean	SD	Range		IQR		Median
				Min	Max	Q1	Q3	
**Width**	Point A	2.53	0.92	1.09	4.06	1.84	2.86	2.32
	Point B	2.92	0.90	0.85	4.71	2.30	3.50	2.76
	Point C	2.61	0.78	0.87	4.08	2.06	3.08	2.63
**Thickness**	Point A	0.86	0.29	0.24	1.44	0.68	1.01	0.82
	Point B	0.52	0.16	0.26	0.89	0.40	0.62	0.50
	Point C	0.69	0.27	0.19	1.29	0.50	0.87	0.68

**Table 3 ijerph-19-05795-t003:** Frequency analysis of Koreans and Europeans according to the type of plantaris tendon.

Type	European (Olewnik, 2017 [15])	Korean (Current Study)	*p*-Value
**1**	22 (44.0)	39 (36.1)	0.849
**2**	9 (18.0)	16 (14.8)	
**3**	4 (8.0)	15 (13.9)	
**4**	2 (4.0)	6 (5.6)	
**5**	11 (22.0)	25 (23.1)	
**Absent**	2 (4.0)	7 (6.5)	
**Total**	50 (100.0)	108 (100.0)	

The data are expressed by number (percent).

## Data Availability

The data presented in this study are available on request from the corresponding author. The data are not publicly available due to donated cadaveric study.

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
