# Peer review of "Anatomical Classification for Plantaris Tendon Insertion and Its Clinical Implications: A Cadaveric Study"

_ijerph, 2022, doi:10.3390/ijerph19105795_

Round 1
Reviewer 1 Report
1-"We suggest PT may be potentially involved in the Achilles tendinopathy with both insertional and mid-portion of CT." Not a direct conclusion derived from the results.
2-"Of the 108 lower limbs, 68 limbs (63%) were from males and 40 limbs (37%) were from females. " Studying both feet of each patient leads to paired-samples effect. It would be appropriate to take one foot of each patient or to run a GEE statistical analysis.
3-How was the sample size arrived at?
4-How was the normality of data tested?
5-ICMJE 4 Authorship criteria require each author to "Substantial contributions to the conception or design of the work; or the acquisition, analysis, or interpretation of data for the work". Please clarify the point.
6-References could be updated and diversified.
Author Response
Reviewer 1
1-"We suggest PT may be potentially involved in the Achilles tendinopathy with both insertional and mid-portion of CT." Not a direct conclusion derived from the results.
Answer: Thanks for the good comment.
I agree with your opinion, and it is believed that the previous conclusions can be a leap forward for the results. Clinical studies have reported that CT (Calcaneal tendon) tendinopathy (Achilles tendinopathy) occurs due to friction caused by the anatomical relationship between PT and CT. Our results on the morphology of PT insertions are thought to provide evidence that they may anatomically affect tendinopathy occurring in CT. Therefore, the conclusion was appropriately revised as follows.
“This study confirmed the five types according to the location of PT and area of its insertion related CT. In addition, the morphology of PT insertion may be anatomically likely to influence the occurrence of tendinopathy in CT.”
2-"Of the 108 lower limbs, 68 limbs (63%) were from males and 40 limbs (37%) were from females. " Studying both feet of each patient leads to paired-samples effect. It would be appropriate to take one foot of each patient or to run a GEE statistical analysis.
Answer: Thanks for the good comment.
After conducting a statistical review by a statistician, the statistical analysis method was appropriately revised. This is described in the Statistical Analysis section as follows.
“Inter-class reliability for all measurements were calculated by interclass correlation coefficient (ICC). Data were analyzed using IBM SPSS Statistics version 23.0 for Windows (IBM Co., Armonk, NY, USA), and a p value less than 0.05 was considered statistically significant. All analyzes were performed using nonparametric methods because our data did not follow the normality assumption. The comparison between ethnicity, gender and side measurements of type classification according to the insertion of PT was analyzed by Generalized Estimating Equation (GEE). The mean comparison of PT width and thickness according to gender was analyzed by Mann-Whitney U test, and the mean comparison between left and right was analyzed by Wilcoxon-sign rank test.”
3-How was the sample size arrived at?
Answer: Thanks for the good comment. This study is an observational study through cadaveric dissection, so statistical sample size derivation was not required. In this study, cadavers that were agreed upon and donated to the medical school for the purpose of education and research were used.
4-How was the normality of data tested?
Answer: Thanks for the good comment.
After conducting a statistical review by a statistician, the statistical analysis method was appropriately revised. This is described in the Statistical Analysis section as follows.
“Inter-class reliability for all measurements were calculated by interclass correlation coefficient (ICC). Data were analyzed using IBM SPSS Statistics version 23.0 for Windows (IBM Co., Armonk, NY, USA), and a p value less than 0.05 was considered statistically significant. All analyzes were performed using nonparametric methods because our data did not follow the normality assumption. The comparison between ethnicity, gender and side measurements of type classification according to the insertion of PT was analyzed by Generalized Estimating Equation (GEE). The mean comparison of PT width and thickness according to gender was analyzed by Mann-Whitney U test, and the mean comparison between left and right was analyzed by Wilcoxon-sign rank test.”
5-ICMJE 4 Authorship criteria require each author to "Substantial contributions to the conception or design of the work; or the acquisition, analysis, or interpretation of data for the work". Please clarify the point.
Answer: Each author has made a significant contribution to the concept or design of the work, or the collection, analysis, or interpretation of data about this study, and has clearly stated the author's contribution to this manuscript as follow.
“Author Contributions: Conceptualization, J.H.P., J.C., K.R.P.; methodology, K.R.P.; formal analysis, K.R.P.; investigation, J.C. K.R.P.; data curation, D.K., H.W.K., M.L., Y.J.C., K.R.P., J.C.; original draft preparation, K.R.P., J.C.; manuscript review and editing, J.H.P., J.C., K.R.P; visualization, K.R.P., K.H.Y.; funding acquisition, J.C.; J.H.P. and J. C. contributed equally to this work.”
6-References could be updated and diversified.
Answer: We have made efforts to update and diversify the references based on your comments.

Reviewer 2 Report
Thank you for the opportunity to review your manuscript, Anatomical classification for plantaris tendon insertion and its clinical implications: A cadaveric study
The purpose of this study is to ascertain the morphological characteristics of the plantaris tendon insertion dissection of Korean cadavers and to classify the types of plantaris tendon insertion related to calcaneal tendon .
It is an anatomical study, necessary for its purpose, with very generous sample size. However, there are aspects that I consider should be improved.
The conclusions of both the abstract and the body of the manuscript should be modified.
It can not reach conclusions such as involvement in Achilles tendinopathy, for which it is not the right design.
It is explained that a classical dissection was performed, but a brief description of the process would be nice.
It is not explained how the thickness was measured and whether the tool used is reliable.
It is said that a reliability study was carried out, but no data from this study, such as SEM and MDD are given.
The 205-220 line is a repetition of results, which should focus on discussing the findings.
Author Response
Reviewer 2
Thank you for the opportunity to review your manuscript, Anatomical classification for plantaris tendon insertion and its clinical implications: A cadaveric study
The purpose of this study is to ascertain the morphological characteristics of the plantaris tendon insertion dissection of Korean cadavers and to classify the types of plantaris tendon insertion related to calcaneal tendon .
It is an anatomical study, necessary for its purpose, with very generous sample size. However, there are aspects that I consider should be improved.
The conclusions of both the abstract and the body of the manuscript should be modified.
It can not reach conclusions such as involvement in Achilles tendinopathy, for which it is not the right design.
Answer: Thanks for the good comment.
I agree with your opinion, and it is believed that the previous conclusions can be a leap forward for the results. Clinical studies have reported that CT (Calcaneal tendon) tendinopathy (Achilles tendinopathy) occurs due to friction caused by the anatomical relationship between PT and CT. Our results on the morphology of PT insertions are thought to provide evidence that they may anatomically affect tendinopathy occurring in CT. Therefore, the conclusion was appropriately revised as follows.
“This study confirmed the five types according to the location of PT and area of its insertion related CT. In addition, the morphology of PT insertion may be anatomically likely to influence the occurrence of tendinopathy in CT.”
It is explained that a classical dissection was performed, but a brief description of the process would be nice.
Answer: Thanks for the good comment.
We agree with your comments that further explanation of the dissection process is necessary.
Therefore, we added the following sentence to "Materials and Methods":
“The skin incision was set with two horizontal incision lines and one vertical incision. The upper horizontal incision line is 10 cm proximal to the line connecting both epicondyle, and the lower horizontal incision line is 5 cm distal to the line connecting the tip of both malleolus. The vertical incision line is a line connecting the middle points of both horizontal incision lines. After skinning, the small saphenous vein and sural nerve were found and removed. The origin of the gastrocnemius and the location of the insertion were checked. The gastrocnemius was cut using surgical scissors at the junction of the medial and lateral heads.”
It is not explained how the thickness was measured and whether the tool used is reliable.
Answer: Thanks for the comment.
We judged that the thickness measurement method was sufficiently explained in "Materials and Methods(118-126)". However, the precision of the tool was confirmed with the manufacturer of the product. Therefore, we added the following sentence to "Materials and Methods":
“All values were measured with an electronic digital caliper (Sincon Corporation, Korea) with an error (precision) of 0.03 mm or less.”
It is said that a reliability study was carried out, but no data from this study, such as SEM and MDD are given.
Answer: Thanks for the comment.
We agree with your comments. Therefore, we presented specific values for inter-class correlation coefficient (ICC) as follows.
“The inter-class correlation coefficient of all measured values for reliability analysis was 0.97 for the width of the PT and 0.96 for the thickness. All measurements were higher than the accepted reliability (0.8) and were employed in the study.”
The 205-220 line is a repetition of results, which should focus on discussing the findings.
Answer: Thanks for the comment.
Lines 205-220 are not repeated descriptions of the results. While reviewing the prior results of classifying PT types through cadaver dissection, we focused on discussing the distribution and proportion of the types by comparing them with each other. And then, our results of this study suggest that there is no statistical difference in the distribution by race.

Reviewer 3 Report
To ascertain the morphological characteristics of the plantaris tendon (PT) insertion, a total of 108 feet from adult formalin-fixed cadavers were dissected. The morphological characteristics and measurements of PT insertion were evaluated and five types of PT insertion types were classified. Then, they concluded that this study classified the morphological characteristics of the PT insertion related to CT and suggested PT may be potentially involved in the Achilles tendinopathy with both insertional and mid-portion of CT.
The authors discussed about the relation among the morphological type, thickness, and width of PT, as a cause of tendinopathy in the discussion section. However, it is difficult to understand the cause for tendinopathy, because the inflammatory change, findings, or the expression of cytokines etc. in the tendon were not presented.
Author Response
Reviewer 3
To ascertain the morphological characteristics of the plantaris tendon (PT) insertion, a total of 108 feet from adult formalin-fixed cadavers were dissected. The morphological characteristics and measurements of PT insertion were evaluated and five types of PT insertion types were classified. Then, they concluded that this study classified the morphological characteristics of the PT insertion related to CT and suggested PT may be potentially involved in the Achilles tendinopathy with both insertional and mid-portion of CT.
The authors discussed about the relation among the morphological type, thickness, and width of PT, as a cause of tendinopathy in the discussion section. However, it is difficult to understand the cause for tendinopathy, because the inflammatory change, findings, or the expression of cytokines etc. in the tendon were not presented.
Answer: Thanks for the good comment.
I agree with your opinion, and it is believed that the previous conclusions can be a leap forward for the results. Clinical studies have reported that CT (Calcaneal tendon) tendinopathy (Achilles tendinopathy) occurs due to friction caused by the anatomical relationship between PT and CT. Our results on the morphology of PT insertions are thought to provide evidence that they may anatomically affect tendinopathy occurring in CT. Therefore, the conclusion was appropriately revised as follows.
“This study confirmed the five types according to the location of PT and area of its insertion related CT. In addition, the morphology of PT insertion may be anatomically likely to influence the occurrence of tendinopathy in CT.”
Also, reflecting your comment, the discussion has been added as follows.
“Sterkenburg et al. [14] reported that PT was closely related to the CT in 11 of 107 study cases. It was suggested that these two tendons generate opposite forces in some ankle joint movements, causing traction to the peritendinous tissue at the level of the mid-portion CT, resulting in Achilles tendinopathy. The mechanism they argued did not consider the tendinous morphological characteristics (changes in thickness and width by location) suggested in this study. However, if you consider the physical movement between the two tendons that are in contact with each other, not only the course or insertion of the PT, but also the change in the shape of the tendon should be treated as important. Also, in healthy normal person or people the Achilles and plantaris tendons, being located in the single paratenon, can glide freely without causing pain [22]. The results of our study show that CT and PT course together with the same paratenon in most cases except for types 3 and 4. This suggests that if PT is thickened in a pathological state, it is highly likely to cause Achilles mid-portion tendinopathy by acting as a compressive or shearing force on CT sufficiently. Additionally, in type 5, the possibility that PT may act on Achilles insertional tendinopathy cannot be excluded. However, since tendinopathy is caused by various factors such as inflammatory changes and tendon cytokine expression, it should not be overlooked that the association between PT and CT should be limited only from an anatomical point of view.”

Round 2
Reviewer 2 Report
The changes presented by the authors are acceptable to me. Thank you.
This manuscript is a resubmission of an earlier submission. The following is a list of the peer review reports and author responses from that submission.
Round 1
Reviewer 1 Report
No comment.
Reviewer 2 Report
The authors presented an adapted classification of the insertion of the plantaris tendon and documented in a cohort of Asian body donors. The strength of the manuscript is a rather large cohort of samples investigated. However, the study is not completely novel, but provides a specific Asian cohort.
Several question are interesting. How about its interrelation to the plantar aponeurosis and Karger`s fat pad? The plantaris muscle tendon could be used for tendon reconstruction as autograft – does it have consequences for patients to excise it?
Line 33 “feet” better to write “legs”
Line 35: remove surplus point
Line 57: “across the posterior of the soleus” please rewrite
Line 125: “medical” means “medial”
Figure 2+3 could be fused together showing in the upper row the scheme and below the examples. Instead of “A” to “E” I would write type 1 to 5.
Figure 2: add the information on which classification e.g. Cummins/Olewnik et al (?) it was based on.
Figure 3 legend: I would assume it is shown as a medial view, but it should be stated in the legend.
Was the difference between absence of PT in male and female individuals significant?
Line 174: add “leg” at the end of the sentence.
Line 260(264: “specific insertion types” exactly which type predisposes to ATT?
Line 280: how can it indeed cause mid-portion tendinopathy?